# First COVID-19 Booster Dose in the General Population: A Systematic Review and Meta-Analysis of Willingness and Its Predictors

**DOI:** 10.3390/vaccines10071097

**Published:** 2022-07-08

**Authors:** Petros Galanis, Irene Vraka, Aglaia Katsiroumpa, Olga Siskou, Olympia Konstantakopoulou, Theodoros Katsoulas, Theodoros Mariolis-Sapsakos, Daphne Kaitelidou

**Affiliations:** 1Clinical Epidemiology Laboratory, Faculty of Nursing, National and Kapodistrian University of Athens, 11527 Athens, Greece; aglaiakat@nurs.uoa.gr; 2Department of Radiology, P. & A. Kyriakou Children’s Hospital, 11527 Athens, Greece; irenevraka@yahoo.gr; 3Department of Tourism Studies, University of Piraeus, 18534 Piraeus, Greece; olsiskou@nurs.uoa.gr; 4Center for Health Services Management and Evaluation, Faculty of Nursing, National and Kapodistrian University of Athens, 11527 Athens, Greece; olykonstant@nurs.uoa.gr (O.K.); dkaitelid@nurs.uoa.gr (D.K.); 5Faculty of Nursing, National and Kapodistrian University of Athens, 11527 Athens, Greece; tkatsoul@nurs.uoa.gr (T.K.); tmariolis@nurs.uoa.gr (T.M.-S.)

**Keywords:** COVID-19, vaccination, booster dose, willingness, refusal, predictors

## Abstract

The emergence of breakthrough infections and new highly contagious variants of SARS-CoV-2 threaten the immunization in individuals who had completed the primary COVID-19 vaccination. This systematic review and meta-analysis investigated, for the first time, acceptance of the first COVID-19 booster dose and its associated factors among fully vaccinated individuals. We followed the PRISMA guidelines. We searched Scopus, Web of Science, Medline, PubMed, ProQuest, CINAHL and medrxiv from inception to 21 May 2022. We found 14 studies including 104,047 fully vaccinated individuals. The prevalence of individuals who intend to accept a booster was 79.0%, while the prevalence of unsure individuals was 12.6%, and the prevalence of individuals that intend to refuse a booster was 14.3%. The main predictors of willingness were older age, flu vaccination in the previous season, and confidence in COVID-19 vaccination. The most important reasons for decline were adverse reactions and discomfort experienced after previous COVID-19 vaccine doses and concerns for serious adverse reactions to COVID-19 booster doses. Considering the burden of COVID-19, a high acceptance rate of booster doses could be critical in controlling the pandemic. Our findings are innovative and could help policymakers to design and implement specific COVID-19 vaccination programs in order to decrease booster vaccine hesitancy.

## 1. Introduction

The emergence of breakthrough infections and new highly contagious variants of SARS-CoV-2 threaten the immunization in individuals who had completed the primary COVID-19 vaccination [1,2]. Moreover, breakthrough infections are more common in immunodeficient patients, causing significant clinical outcomes, e.g., hospitalization and mortality [3,4]. Importantly, evidence shows that new SARS-CoV-2 variants of concern may have reduced COVID-19 vaccine susceptibility and have increased infectivity [5,6].

COVID-19 vaccines were administrated by early 2021, were proven sufficiently safe and effective and demonstrated high protection against SARS-CoV-2 infections, hospitalizations and deaths [7,8]. However, efficacy or effectiveness against SARS-CoV-2 infection and symptomatic disease decreased six months after full vaccination [9]. Thus, there has been a discussion about the need for a COVID-19 booster dose [10,11]. Eventually, several countries deployed extra COVID-19 vaccine doses, especially for vulnerable groups, in order to strengthen immune responses and prolong protection against SARS-CoV-2.

Primary real-world data support the effectiveness of a first booster dose since SARS-CoV-2 infection rate, hospitalization rate and COVID-19-related mortality are lower among individuals who receive a first booster shot after the primary vaccination [12,13]. In addition, a first booster dose, when it is administered several months after the second COVID-19 vaccine dose, induces a robust immune response and prolongs protection [14,15]. Moreover, many countries have already recommended a second booster dose for high-risk groups to further support immunization against SARS-CoV-2. Thus, public willingness to accept booster doses could be a viable option to shore up protection against COVID-19 and control the pandemic.

Moreover, adherence to the recommended protective measures during the COVID-19 pandemic (e.g., COVID-19 vaccination) is influenced by several factors, such as knowledge and attitudes, especially among older adults [16,17]. In addition, higher levels of knowledge concerning COVID-19 are associated with COVID-19 vaccine acceptance [18,19]. Adequate knowledge about COVID-19 vaccination and positive beliefs regarding COVID-19 strengthen the confidence of the population in proactive behaviors to prevent COVID-19.

Until now, no systematic review has investigated individuals’ intention to accept COVID-19 booster doses. Therefore, we conducted a systematic review and meta-analysis to investigate the acceptance of a first booster dose and its associated factors among fully vaccinated individuals. Moreover, we estimated the percentage of the fully vaccinated individuals who refused a first booster dose, as well as those who were unsure.

## 2. Materials and Methods

### 2.1. Data Sources and Strategy

We applied the Preferred Reporting Items for Systematic Reviews and Meta-Analysis (PRISMA) guidelines for this systematic review and meta-analysis [20]. We searched Scopus, Web of Science, Medline, PubMed, ProQuest, CINAHL and a pre-print service (medrxiv) from inception to 21 May 2022. We used the following strategy in all fields: (((vaccin*) AND (COVID-19)) AND (SARS-CoV-2)) AND (booster).

### 2.2. Selection and Eligibility Criteria

We used the following inclusion criteria: studies reporting the willingness of fully vaccinated individuals to receive or refuse a first COVID-19 booster dose, quantitative studies, studies that included adults, studies with samples from the general population, studies published in English and studies published in peer-reviewed journals. We excluded studies involving specific population groups (e.g., healthcare workers and patients) since the aim of our review was to investigate acceptance of a first COVID-19 booster dose among fully vaccinated individuals in the general population only. We considered that specific population groups should not be mixed with the general population since they are different populations regarding their attitudes toward COVID-19 vaccination. Moreover, we excluded reviews, protocols, case reports, opinion articles, commentaries, editorials and letters to the editor.

We used Zotero software to remove duplications, and we then consecutively reviewed titles, abstracts and full texts. Moreover, we hand-searched the reference lists of all relevant reviews and articles. Two independent authors performed study selection, and the most experienced authors resolved the discrepancies.

### 2.3. Data Extraction

Two authors independently collected data for the following items: reference, country, data collection time, sample size, percentage of females, age of participants, study design, sampling method, recruitment method, response rate, publication type (journal or pre-print), question to measure public willingness to accept a first COVID-19 booster dose, response scales, percentage of individuals who intended to accept a booster dose, percentage of individuals who intended to refuse a booster dose, percentage of individuals who were unsure and predictors of individuals’ decision to accept a booster dose.

### 2.4. Quality Assessment

Two independent authors used the Joanna Briggs Institute critical appraisal tool to assess the risk of bias of the included studies [21]. The most experienced authors solved any discrepancies. The Joanna Briggs Institute critical appraisal tool consists of eight questions (e.g., Were the criteria for inclusion in the sample clearly defined? Were the study subjects and the setting described in detail?, etc.). Response options for each question were the following: “yes”, “no”, “unclear” and “not applicable”. Considering the answers to the eight questions, we divided studies into low, medium or high risk of bias.

### 2.5. Statistical Analysis

There was great variation in the way that authors of different studies measured willingness and refusal of individuals towards a first COVID-19 booster dose. In particular, authors used different questions (e.g., “How likely do you think you are to get a COVID-19 booster vaccine if/when you are offered one?”, “Are you willing to receive the potential additional dose of the COVID-19 vaccine if it would be made available?”, “Do you accept receiving the COVID-19 booster vaccine?”, etc.) and different answers. Possible response options were in (a) yes/no options, (b) yes/no/unsure options and (c) Likert scales (e.g., very willing; willing; fair; unwilling; very unwilling; unsure). For each study, we followed the authors’ criteria regarding the decision of participants to accept a booster dose or not. Then, we divided the positive answers of participants by the total number of participants in order to calculate the percentage of participants who intended to accept a first COVID-19 booster dose. In the same way, we calculated the percentage of participants who intended to refuse a booster dose and the percentage of participants who were unsure. Finally, we used the Freeman–Tukey Double Arcsine method to transform the above three percentages. Moreover, we calculated the 95% confidence intervals (CI) for these percentages [22].

We performed a meta-analysis to calculate the prevalence of each included study, as well as the pooled estimate. We used the inconsistency index I2 to assess statistical heterogeneity among studies, with values higher than 75% indicating high heterogeneity [23]. We adopted a random effect model for all analyses since the statistical heterogeneity was very high [23]. We pre-specified the following sources of heterogeneity: data collection time, sample size, gender distribution, age, study design, sampling method, recruitment method, response rate, publication type (journal or pre-print service), response scales (studies with or without unsure option), quality of the studies and the country in which studies were conducted. Due to limited data for some variables (response rate), limited variability in some variables (study design, recruitment method, studies quality and publication type) and high heterogeneity in the measurement of some variables (age), we decided to perform subgroup analyses for the sampling method, response scales and countries. Moreover, we used data collection time, sample size and gender distribution as independent variables in meta-regression models. We considered data collection time as a continuous variable, assigning the number 1 for studies that were conducted in April 2021, the number 2 for studies that were conducted in May 2021, etc. A leave-one-out sensitivity analysis was used to determine the influence of each study on the overall prevalence. Funnel plots and the Egger’s test (*p*-value < 0.05) were used to assess potential publication bias [24]. The analysis was performed using OpenMeta[Analyst] statistical software [25].

High heterogeneity in the way that authors investigated the relationship between independent variables and individuals’ willingness to accept a first COVID-19 booster dose did not allow the performance of a meta-analysis. However, in order to quantify the magnitude of independent variables, we measured the percentage of studies finding positive or negative significant relationships (*p*-value < 0.05). Therefore, we calculated this percentage by dividing the number of studies with a positive significant relationship between the independent variable and individuals’ willingness to accept a booster dose by the total number of studies that examined the independent variable.

## 3. Results

### 3.1. Identification and Selection of Studies

Initially, we found 4712 unique records. Applying the inclusion and exclusion criteria, we identified 14 articles (Figure 1).

### 3.2. Characteristics of the Studies

Details of the studies included in this systematic review are presented in Table 1. We found 14 studies including 104,047 fully vaccinated individuals [26,27,28,29,30,31,32,33,34,35,36,37,38,39]. Nine studies were conducted in Asia (four in China; two in Jordan; one in Vietnam; one in Japan; one in Indonesia), three studies in Europe (one in the United Kingdom; one in Denmark; one in Poland), one study in the USA and one study in Africa (Algeria). Data collection times among studies ranged from April 2021 to March 2022. Sample sizes ranged from 413 to 31,721 individuals, with a median number of 2237 individuals. The percentage of females was higher than the percentage of males in seven studies, while in one study, we found the opposite. In addition, in five studies, the distribution of the two genders was almost equal. Thirteen studies were cross-sectional, and one study was cohort. Data collection was performed through online questionnaires in all studies. Seven studies used a convenience sample, six studies used the snowball sampling method and one study used a stratified random sample. All studies were published in peer-reviewed journals. Seven studies included an “unsure” response option for individuals’ willingness to accept a first COVID-19 booster dose, while seven studies did not include this response option (Table 2).

### 3.3. Characteristics of the Studies

The risk of bias was low in thirteen studies and moderate in one study. The quality assessment of the studies included in this review is shown in Appendix A.

### 3.4. Individuals’ Willingness and Refusal to Accept a First COVID-19 Booster Dose

Fourteen studies reported the number of individuals who intend to accept a first COVID-19 booster dose. The prevalence of individuals who intend to accept a booster was 79.0% (95% CI: 71.8–85.3%) (Figure 2). The heterogeneity between results was very high (I^2^ = 99.84%, *p*-value for the Hedges Q statistic < 0.001). Individuals’ willingness to accept a booster dose ranged from 44.6% to 97.9%. A leave-one-out sensitivity analysis showed that no single study had a disproportional effect on the overall willingness, which varied between 77.0% (95% CI: 69.3–83.9%) and 81.4% (95% CI: 74.6–87.3%). Publication bias was probable since the *p*-value for the Egger’s test was lower than 0.05, and the funnel plot was asymmetrical (Appendix A).

Thirteen studies reported the number of individuals who intend to refuse a booster. The prevalence of individuals who intend to refuse a booster was 14.3% (95% CI: 8.4–21.4%) (Figure 3). The heterogeneity between results was very high (I^2^ = 99.82%, *p*-value for the Hedges Q statistic < 0.001). Individuals’ refusal to accept a booster dose ranged from 1.0% to 43.7%. A leave-one-out sensitivity analysis showed that no single study had a disproportional effect on the overall refusal, which varied between 12.0% (95% CI: 7.4–17.4%) and 15.8% (95% CI: 9.1–23.9%). Publication bias was probable since the *p*-value for the Egger’s test was lower than 0.05, and the funnel plot was asymmetrical (Appendix A).

Six studies presented the number of individuals reporting that they were unsure of their intention to accept a booster. The prevalence of unsure individuals was 12.6% (95% CI: 6.8–19.9%) (Figure 4). The heterogeneity between results was very high (I^2^ = 99.80%, *p*-value for the Hedges Q statistic < 0.001). The prevalence of unsure individuals ranged from 4.3% to 24.7%. A leave-one-out sensitivity analysis showed that no single study had a disproportional effect on the overall refusal, which varied between 10.2% (95% CI: 4.5–17.5%) and 14.2% (95% CI: 7.1–23.1%). Publication bias was probable since the *p*-value for the Egger’s test was lower than 0.05, and the funnel plot was asymmetrical (Appendix A).

### 3.5. Impact of the “Unsure” Response Option

The prevalence of willingness was lower in studies with an “unsure” response option (69.6%, 95% CI = 63.3–75.9%, I^2^ = 99.82%) than in studies without an “unsure” response option (83.3%, 95% CI = 76.5–90.1%, I^2^ = 99.76%). On the other hand, there was no impact of the “unsure” response option on the individuals’ refusal of a COVID-19 booster dose. In particular, the prevalence of refusal in studies with an “unsure” response option was 16.4% (95% CI = 12.5–20.4%, I^2^ = 99.73%), and in studies without an “unsure” response option, it was 16.7% (95% CI = 9.9–23.5%, I^2^ = 99.76%).

### 3.6. Impact of the Sampling Method

The sampling method did not affect the prevalence of individuals’ willingness to accept a booster dose. In particular, the prevalence of willingness in studies with a convenience sample was 77.6% (95% CI = 60.8–90.7%, I^2^ = 99.49%), and in studies that adopted a snowball sampling method, it was 76.9% (95% CI = 63.3–88.1%, I^2^ = 99.84%). Similarly, the impact of the sampling method on the prevalence of individuals’ refusal to accept a booster was very low, since the prevalence of refusal in studies with a convenience sample was 14.8% (95% CI = 3.7–31.6%, I^2^ = 99.86%), and in studies that adopted a snowball sampling method, it was 16.1% (95% CI = 3.5–35.2%, I^2^ = 99.83%).

### 3.7. Impact of the Countries

We separated countries according to the Confucian culture circle (China, Vietnam and Japan), and we found that the prevalence of willingness was higher in studies that were conducted in countries that belong to the Confucian culture circle (88.5%, 95% CI = 79.2–95.3%, I^2^ = 99.78%) than in studies that were conducted in countries that do not belong to the Confucian culture circle (71.8%, 95% CI = 57.5–84.1%, I^2^ = 99.89%). On the other hand, there was no impact of the countries on the individuals’ refusal of a COVID-19 booster dose. In particular, the prevalence of refusal in studies that were conducted in countries that belong to the Confucian culture circle was 7.8% (95% CI = 1.3–19.2%, I^2^ = 99.72%), and in studies that were conducted in countries that do not belong to the Confucian culture circle, it was 7.0% (95% CI = 3.3–11.9%, I^2^ = 99.70%).

### 3.8. Meta-Regression Analysis

An increased sample size was associated with an increase in individuals’ willingness to accept a booster dose (coefficient = 0.000012, 95% CI = 0.000003 to 0.00021, *p* = 0.012). Moreover, an increased percentage of females in studies was associated with a decreased willingness (coefficient = −0.930, 95% CI = −1.736 to −0.123, *p* = 0.024). On the other hand, data collection time did not affect willingness (coefficient = -0.008, 95% CI = −0.055 to 0.040, *p* = 0.752).

Regarding individuals’ refusal to accept a booster dose, an increased sample size was associated with decreased refusal (coefficient = −0.00001, 95% CI = −0.00002 to −0.000008, *p* = 0.045). The percentage of females (coefficient = 0.101, 95% CI = −0.995 to 1.197, *p* = 0.857) and data collection time (coefficient = −0.008, 95% CI = −0.050 to 0.022, *p* = 0.718) had no impact on the prevalence of refusal.

### 3.9. Predictors of Individuals’ Willingness to Accept a First COVID-19 Booster Dose

Twelve studies investigated predictors of individuals’ willingness to accept a first COVID-19 booster dose (Table 3). Only one study did not use multivariable analysis to eliminate confounders.

Authors have mainly investigated the effect of socio-demographic characteristics on individuals’ willingness to accept further vaccination(s). In particular, in eight out of twelve studies, older individuals were more likely to accept a first COVID-19 booster dose; in two out of twelve studies, younger individuals were more willing; and two studies found no effect with age. Two studies out of ten found that males intend to accept a booster dose more often than females, but two studies found the opposite, and six studies found no effect of gender. Higher educational level was associated with individuals’ willingness to accept a booster in four out of ten studies, while three studies found the opposite relationship, and three studies found no effect of educational level. In two out of four studies, individuals with higher income were more likely to report an intention to accept a booster dose than individuals with lower income, while two studies did not find a significant relationship. Chronic comorbidities presented a controversial issue, since in three out of six studies, individuals with at least one chronic condition were more likely to accept further vaccination; in two out of six studies, chronic comorbidities showed no association with willingness; and in one study, individuals with comorbidity were less likely to accept vaccination. Ethnicity, marital status and residence were non-significant predictors in two out of two, four out of five and eight out of eight studies, respectively.

Few studies investigated the impact of COVID-19-related variables on individuals’ decision to accept a first COVID-19 booster dose. In particular, history of COVID-19 infection (three out of six studies; no association in three studies), higher risk perception of infection (three out of five studies; no association in two studies), and higher severity perception of COVID-19 (three out of six studies; negative association in one study; no association in two studies) were associated with an increase in individuals’ willingness to accept a first COVID-19 booster dose.

Confidence in COVID-19 vaccines (two out of two studies) and COVID-19 booster doses (four out of four studies) was associated with an increased acceptance of a first COVID-19 booster dose. On the other hand, adverse reactions and discomfort experienced after previous COVID-19 vaccines doses (four out of four studies) and concerns for serious adverse reactions to COVID-19 booster doses (two out of two studies) decreased individuals’ willingness to accept a booster.

A limited number of studies identified that a flu vaccination in the previous season (two out of three studies), confidence in the healthcare system/government/physicians (two out of three studies) and compliance with prevention measures (one out of one study) positively affected individuals’ intention to accept a booster dose.

## 4. Discussion

To the best of our knowledge, this is the first systematic review that estimates the acceptance and refusal of a first COVID-19 booster dose among fully vaccinated individuals. Moreover, we reviewed the available literature on predictors of individuals’ willingness to accept a booster dose. Applying inclusion and exclusion criteria, we found 14 studies carried out mainly in Asia from April 2021 to March 2022.

### 4.1. Willingness and Refusal of Individuals to Accept a First COVID-19 Booster Dose

Worldwide, among individuals who are fully vaccinated, 79% would take a first booster if recommended. There is great variability in the willingness rate among studies, from 44.6% to 97.9%. Moreover, we found that 14.3% of fully vaccinated individuals would refuse a booster dose, and 12.6% are unsure whether they would accept a booster dose. Thus, even among fully vaccinated individuals, many of them have no fixed opinion on boosters and may yet be persuadable. A similar review including studies in the general population found lower global primary COVID-19 vaccination willingness (66%) [40]. The willingness rate is even lower among healthcare workers (63.5%) [41] and parents who intend to vaccinate their children (60.1%) [42]. The acceptance rate of the first COVID-19 booster dose is higher than that of primary doses, but there is still a potential for improvement. Moreover, considering that this review included studies with fully vaccinated individuals, the acceptance rate of vaccination is relatively low. Our review shows that, even among those individuals who chose to receive the primary COVID-19 doses, a significant percentage expressed substantial hesitancy or reluctance to accept a booster dose. Concerns about booster safety, effectiveness and side effects, the belief that the primary COVID-19 vaccination provides sufficient immunization and a low severity perception of COVID-19 seem to be the primary reasons for booster vaccine hesitancy [25,26,30,34,35]. New highly contagious variants of SARS-CoV-2 and the potential waning of vaccine protection highlight the urgent need for booster vaccination in order to end the COVID-19 pandemic. Therefore, policymakers should develop and implement effective communication strategies, emphasizing vaccination safety and effectiveness, in order to increase the proportion of individuals receiving the booster doses.

### 4.2. Subgroup Analyses

Interestingly, we found that the willingness rate was lower in studies with an “unsure” response option than in studies without an “unsure” response option. This finding is confirmed by two similar meta-analyses including individuals from the general population and parents [42,43]. The public willingness rate of primary COVID-19 vaccination in studies with an “unsure” response option was 63.5%, while in studies without an “unsure” response option, it was 82.8% [43]. Similarly, the proportion of parents who intend to vaccinate their children against COVID-19 in studies with an “unsure” response option was 58.3%, while in studies without an “unsure” response option, it was 64.5% [42]. It is reasonable that an “unsure” response option decreases the acceptance rate of booster doses since many individuals are unsure whether they would accept a booster.

Our meta-analysis identified that the increased percentage of females in studies was associated with decreased willingness. This finding is confirmed by a recent meta-analysis that found lower COVID-19 vaccination intentions among females than males [44]. Moreover, females are less likely than males to accept a first COVID-19 booster dose [31,34]. Moreover, more males than females actually receive a COVID-19 vaccine [45]. Females with limited knowledge regarding pregnancy, fertility and breastfeeding could explain their hesitancy towards COVID-19 vaccines [46,47]. Our finding is also in line with previous research on other vaccines. For example, male adolescents had a higher likelihood of being fully vaccinated compared with female adolescents [48]. In addition, females have lower vaccination rates than males in the case of pandemic influenza and influenza vaccinations [49,50,51]. Psychological and hormonal gender differences could explain vaccine hesitancy among females [52,53].

Moreover, we found that the prevalence of willingness was higher in studies that were conducted in countries that belong to the Confucian culture circle (China, Vietnam and Japan) than in studies that were conducted in other countries. The cultural background of individuals seems to influence their decision to accept a booster dose. However, there is a need to conduct studies that directly compare individuals from different cultural backgrounds in order to find valid results.

### 4.3. Predictors of Individuals’ Willingness to Accept a First COVID-19 Booster Dose

According to our review, positive attitudes and perceptions, including confidence in COVID-19 vaccination, the healthcare system, government and physicians, was associated with a willingness to receive a booster dose. On the other hand, negative attitudes and perceptions, including perceptions of the adverse reactions and low levels of safety and effectiveness of COVID-19 vaccination, were significantly associated with hesitance and reluctance. These findings are confirmed by similar reviews since individuals with positive attitudes towards COVID-19 vaccination are more likely to accept and less likely to refuse a vaccine [40,54,55]. Many people discuss concerns about the safety, efficacy and effectiveness of COVID-19 vaccines and how they are less willing to vaccinate. Public concerns about possible side effects of COVID-19 vaccines play a critical role in the intention to vaccinate. Evidence shows that the acceptance rate of COVID-19 vaccines changes with reported vaccine effectiveness changes [56,57]. Increased COVID-19 vaccine efficacy and prolonged protection time are associated with an increased acceptance rate [58]. Moreover, public attitudes towards COVID-19 vaccination tend to change over time because people want to see more data that have been gathered after vaccination [59,60]. In addition, vulnerable groups (e.g., elderly and patients who are more susceptible to clinical complications) are less willing to be vaccinated because they are more concerned about the side effects [61,62]. Therefore, reliable information regarding the safety and effectiveness of COVID-19 vaccines should be provided to those who have previously experienced side effects. Moreover, it is an unprecedented challenge for public health authorities to achieve a valid post-marketing surveillance in order to determine the long-term side effects of COVID-19 vaccinations [63]. This continuous evidence on COVID-19 vaccine safety may build trust among the general population and public health authorities to increase the vaccine confidence level.

We found that several socio-demographic characteristics affect individuals’ decision to accept a booster dose. In particular, older individuals were more likely to accept a booster. Evidence supports this finding since increased age is associated with increased COVID-19 vaccination intentions and uptake [36,41,50]. The elderly may have a greater sense of responsibility and accountability for themselves and their societies relative to younger individuals. Moreover, older people may think they are more susceptible to COVID-19-related clinical complications since an older age is a predictor of mortality in COVID-19 patients [64,65]. On the other hand, a lower risk perception of COVID-19 infection among younger people could explain their vaccine hesitancy [54].

Our review identified that people with a higher educational level were more likely to accept a booster dose. This finding is confirmed by the literature since an increased educational level is associated with an increased primary COVID-19 vaccine acceptance rate [36,50,51]. Well-educated individuals have a higher risk perception of COVID-19 infection, while less-educated people are vaccine-hesitant because they do not think they will be affected by COVID-19 [66]. Moreover, individuals who have a higher level of education also have access to multiple and valid information sources [67]. An educational level affects the general knowledge and awareness of individuals towards COVID-19 vaccines, while conspiracy theories, fake news and misinformation increase COVID-19 vaccine hesitancy, especially among less-educated people. Thus, there is a need for the implementation of social campaigns that deliver trusted news regarding COVID-19 vaccination.

Additionally, we found that people with higher income are more likely to accept a booster dose than those with lower income. Income has an effect on information achievements since people whose incomes are high are more aware of the negative consequences of COVID-19 and more willing to be vaccinated in order to maintain their health [68]. In contrast, people with low incomes are less likely to vaccinate since they lack the health insurance and financial resources that may be necessary for access to COVID-19 vaccines [69]. Moreover, socioeconomic disadvantages and health inequalities can increase vaccine hesitancy among low-income people [54]. On the other hand, high-income people have a higher risk perception of COVID-19 infection and are aware of the vaccines’ safety and effectiveness, which help them accept the vaccine more easily [66].

### 4.4. Limitations

Firstly, vaccine acceptance and hesitancy repeatedly change over time as new evidence and vaccination data are observed. Therefore, the findings of this review and meta-analysis may be invalid after a certain period. Longitudinal studies should be conducted in order to clarify individuals’ attitudes towards COVID-19 vaccination and identify changes in preference over time. Secondly, according to our review, only one study used a stratified random sample, while the other studies used convenience and snowball sampling methods. Sampling methods may influence vaccination willingness and refusal, and our findings may have been different if sampling had been different. For example, it is difficult to reach minority and under-represented groups through convenience and snowball sampling methods. Therefore, future studies should adopt random sampling in order to achieve an unbiased representation of the source populations. Additionally, 12 out of 14 studies in our review were conducted in Asian and European countries. This over-presentation of Asian and European countries may introduce bias in our review, and thus, our findings could not be generalized in other continents. Thus, there is an urgent need for further studies in continents other than Asia and Europe. Moreover, almost all studies in our review (13 out of 14) were cross-sectional, and thus, causal inferences could not be achieved. Prospective cohort studies should be conducted in order to reduce bias and achieve more valid results. Similarly, only 12 studies investigated predictors of individuals’ willingness to accept a first COVID-19 booster dose. Even worse, most studies focused only on the effect of socio-demographic characteristics on individuals’ willingness to accept booster doses. Future research should expand our knowledge regarding the predictors of COVID-19 willingness. For example, we could examine the role of psychological factors, social media variables, infection-related variables, etc. Moreover, we did not include specific population groups (e.g., healthcare workers and patients) in our review since our aim was to investigate the acceptance of a first COVID-19 booster dose in the general population only. We consider that patients have a different approach regarding COVID-19 vaccination since it is well-known that comorbidities significantly increase the negative outcomes of COVID-19, and thus, vaccination acceptance among patients is expected to be higher than the general population. In addition, healthcare workers are a high-risk group, and the frame for their decision to accept COVID-19 vaccines is different than that of the general population. For example, COVID-19 vaccination is obligatory for healthcare workers in nursing homes. Therefore, a systematic review that included specific population groups could add invaluable evidence in this field. Another limitation of our study is that we could not perform subgroup analysis for all the variables that we pre-planned due to limited data and variability and high heterogeneity in the measurements. Future studies should provide more data in order to give us the opportunity to make a better assessment of sources of heterogeneity. Finally, COVID-19 booster uptake should be investigated in the future since intentions do not always predict individuals’ actions. Thus, we should estimate the prevalence of the COVID-19 booster uptake in the general population and the factors that affect this behavior.

## 5. Conclusions

Our findings show that, even among fully vaccinated individuals, many of them have no fixed opinion on booster doses. With a low level of willingness towards receiving booster doses, it may be extremely difficult to manage the COVID-19 pandemic. The situation is becoming even worse due to the waning immunity to the first COVID-19 booster dose and the emergence of highly contagious variants of SARS-CoV-2. Thus, several countries have already recommended a second booster for vulnerable groups [15,70].

Policymakers should emphasize their strategies on COVID-19 vaccine/vaccination-related knowledge since several studies found a positive relationship between this knowledge and vaccine acceptance among adults [18,71,72]. Moreover, COVID-19-related knowledge is associated with the correct practice of preventive measures (e.g., hand washing, facial masks, vaccination, etc.) among the elderly [17,73,74]. In addition, knowledge about COVID-19 is associated with positive vaccine attitudes among healthcare workers since healthcare workers with a higher knowledge level about COVID-19 vaccination are more willing to be vaccinated [75,76]. An important determinant of the acceptance of COVID-19 vaccination among adults with chronic illnesses is knowledge of COVID-19 and information about the COVID-19 vaccine [77,78]. Informative communication with specific population groups and policymakers about COVID-19 vaccination is crucial to decrease vaccine hesitancy.

We should create awareness towards the COVID-19 vaccine in order to increase the acceptance of the COVID-19 vaccine among the general population. In that case, good knowledge about vaccination will help individuals to understand the safety, efficacy and effectiveness of the COVID-19 vaccine. Since several factors affect individuals’ decision to accept a COVID-19 vaccine, a holistic educational approach to improve confidence in the COVID-19 vaccine should be implemented. Moreover, policymakers should develop and implement targeted education for people with a low level of knowledge and a low level of willingness. In that case, transparent procedures, such as vaccination guarantee policies, open vaccine review procedures and reduced vaccine costs, could help to improve the acceptance rate of vaccination. Moreover, mass media and social media messages could promote adherence to protective measures, such as vaccination, hand washing, facial masks, etc.

The decline of individuals to accept future booster doses or even a new COVID-19 vaccine may undermine the public health advantages of a safe and effective vaccine. Therefore, understanding individuals’ willingness to take booster doses and the possible predictors affecting their vaccine attitudes will give policymakers the opportunity to develop and implement effective vaccination programs. Likewise, understanding COVID-19 vaccine hesitancy could improve COVID-19 vaccine uptake and tackle the COVID-19 pandemic. Booster vaccination is a high-priority task, and we should improve attitudes towards COVID-19 booster doses, providing community health education as soon as possible. There is an urgent need for continuous and updated evidence regarding public attitudes towards COVID-19 booster doses since a second booster dose for the general population in autumn 2022 is a possible scenario.

## Figures and Tables

**Figure 1 vaccines-10-01097-f001:**
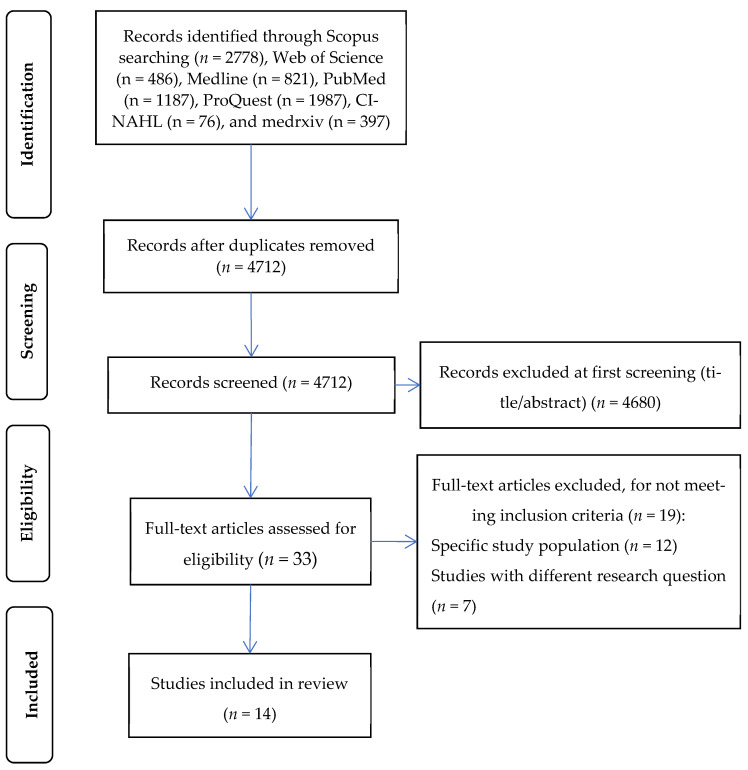
Flowchart of the literature search according to the Preferred Reporting Items for Systematic Reviews and Meta-Analysis.

**Figure 2 vaccines-10-01097-f002:**
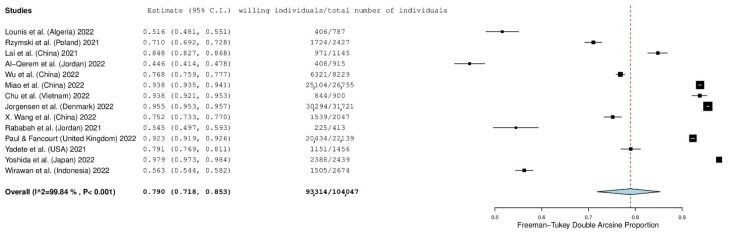
Forest plot of individuals’ willingness to accept a first COVID-19 booster dose. The size of black squares denotes the sample size, while the horizontal line through the square denotes the 95% confidence interval for the willingness proportion.

**Figure 3 vaccines-10-01097-f003:**
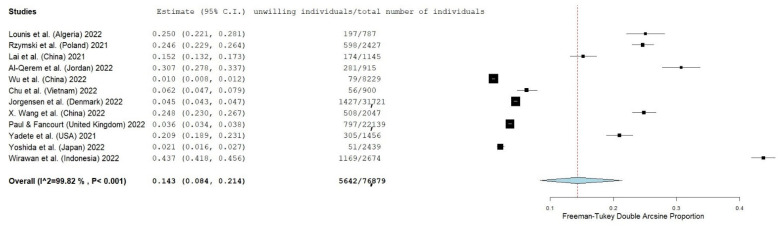
Forest plot of individuals’ refusal to accept a first COVID-19 booster dose. The size of black squares denotes the sample size, while the horizontal line through the square denotes the 95% confidence interval for the refusal proportion.

**Figure 4 vaccines-10-01097-f004:**
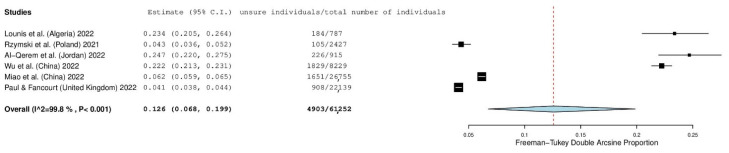
Forest plot of individuals reporting that they were unsure of their intention to accept a first COVID-19 booster dose. The size of black squares denotes the sample size, while the horizontal line through the square denotes the 95% confidence interval for the unsure proportion.

**Table 1 vaccines-10-01097-t001:** Overview of the studies included in this systematic review.

Reference	Country	Data Collection Time	Sample Size (N)	Females (%)	Age, Mean (Standard Deviation)	Study Design	Sampling Method	Recruitment Method	Response Rate (%)	Published In
Lounis et al. [30]	Algeria	January to March 2022	787	61.6	18–40 years, 58.2%; 41–60 years, 36.8%; >60 years, 5%	Cross-sectional	Snowball	Online survey	NA	Journal
Rzymski et al. [34]	Poland	September 2021	2427	50.7	<50 years, 62.3%; ≥50 years, 37.7%	Cross-sectional	Snowball	Online survey	NA	Journal
Lai et al. [29]	China	June 2021	1145	50.3	≤40 years, 65.7%; 41–59 years, 34.3%	Cross-sectional	Snowball	Online survey	NA	Journal
Al-Qerem et al. [26]	Jordan	October to December 2021	915	NR	18–29 years, 45.7%; ≥30 years, 54.3%	Cross-sectional	Convenience	Online survey	NA	Journal
Wu et al. [37]	China	October 2021	8229	69.0	26–45 years, 78.5%; ≥46 years, 21.5%	Cross-sectional	Snowball	Online survey	NA	Journal
Miao et al. [31]	China	August 2021	26,755	52.6	<40 years, 83.8%; ≥40 years, 16.2%	Cross-sectional	Snowball	Online survey	NA	Journal
Chu et al. [27]	Vietnam	November 2021	900	25.7	18–44 years, 91.8%; ≥45 years, 8.2%	Cross-sectional	Convenience	Online survey	NA	Journal
Jørgensen et al. [28]	Denmark	December 2021	31,721	NR	NR	Cross-sectional	Stratified random	Online survey	25	Journal
Wang et al. [35]	China	April to May 2021	2047	59.3	18–44 years, 64.7%; ≥45 years, 35.3%	Cross-sectional	Snowball	Online survey	NA	Journal
Rababa’h et al. [33]	Jordan	August 2021	413	76	18–39 years, 74.6%; ≥40 years, 25.4%	Cross-sectional	Convenience	Online survey	NA	Journal
Paul et al. [32]	United Kingdom	November to December 2021	22,139	51	18–44 years, 44%; ≥45 years, 56%	Cross-sectional	Convenience	Online survey	NA	Journal
Yadete et al. [38]	USA	July 2021	1456	49.7	NR	Cross-sectional	Convenience	Online survey	NA	Journal
Yoshida et al. [39]	Japan	September to October 2021	2439	58.3	52.6 (19.3)	Cohort	Convenience	Online survey	NR	Journal
Wirawan et al. [36]	Indonesia	December 2021 to January 2022	2674	58	29 (24–35) ^a^	Cross-sectional	Convenience	Online survey	NA	Journal

NA, not applicable; NR, not reported. ^a^ median (interquartile range).

**Table 2 vaccines-10-01097-t002:** Response scales and results of individuals’ willingness to accept a first COVID-19 booster dose in studies included in systematic review.

Reference	Question/Statement to Measure Patients’ Willingness	Response Scale ^a^	Willingness Results (%)
Lounis et al. [30]	Are you willing to receive the potential additional dose of the COVID-19 vaccine if it would be made available?	Yes (Y), unsure (U), no (N)	Yes: 51.6 Unsure: 23.4 No: 25
Rzymski et al. [34]	Are you willing to receive the potential additional dose of the COVID-19 vaccine if it would be made available?	Yes (Y), unsure (U), no (N)	Yes: 71.0 Unsure: 4.3 No: 24.7
Lai et al. [29]	If a COVID-19 booster is recommended as a supplement to the current vaccination schedule, would you accept it?	Yes (Y), no (N)	Yes: 84.8 No: 15.2
Al-Qerem et al. [26]	Are you willing to take the booster dose?	Yes (Y), unsure (U), no (N)	Yes: 44.6 Unsure: 24.7 No: 30.7
Wu et al. [37]	To what extent do you want to take the booster COVD-19 vaccine?	Yes, definitely (Y), unsure but tend to be willing (U), unsure but tend to be unwilling (U), definitely no (N)	Yes: 76.8 Unsure: 22.2 No: 1.0
Miao et al. [31]	Are you willing to take the booster dose?	Very willing (Y), willing (Y), fair (U), unwilling (U), very unwilling (U), don’t know (U)	Yes: 93.8 Unsure: 6.2
Chu et al. [27]	If an extra dose of COVID-19 vaccine is available, would you get it?	Definitely yes (Y), probably yes (Y), probably no (N), definitely no (N)	Yes: 93.7 No: 6.3
Jørgensen et al. [28]	Will you accept the booster vaccine?	I have received the booster dose (Y), I have not yet received the invitation to the booster dose, but I wish to be vaccinated with the booster dose (Y), I have received the invitation to the booster dose, and I wish to be vaccinated, but have not yet been vaccinated with the booster dose (Y), I have received an invitation to the booster dose, but a do not wish the booster dose (N), I have not yet received the invitation to the booster dose, and I do not wish to be vaccinated with the booster dose (N), Do not want to answer (N)	Yes: 95.5 No: 4.5
Wang et al. [35]	Are you willing to receive the potential additional dose of the COVID-19 vaccine if it would be made available?	Yes (Y), no (N)	Yes: 75.2 No: 24.8
Rababa’h et al. [33]	Do you accept receiving the COVID-19 booster vaccine?	Yes (Y), unsure (U), no (N)	Yes: 54.5 Unsure: NR No: NR
Paul et al. [32]	How likely do you think you are to get a COVID-19 booster vaccine if/when you are offered one?	A scale from 1 (very unlikely) to 6 (very likely); 5–6 (Y), 3–4 (U), 1–2 (N)	Yes: 92.3 Unsure: 4.1 No: 3.6
Yadete et al. [38]	Do you accept receiving the COVID-19 booster vaccine?	Yes (Y), no (N)	Yes: 79.1 No: 20.9
Yoshida et al. [39]	Do you accept receiving the COVID-19 booster vaccine?	Yes (Y), no (N)	Yes: 97.9 No: 2.1
Wirawan et al. [36]	Will you accept the booster vaccine?	I have received the booster dose (Y), A scale from 1 (would not accept) to 5 (certainly would accept); 5 (Y), 1–4 (N)	Yes: 56.3 No: 43.7

NR: not reported. ^a^ (Y), (N) and (U) indicate extracted response options representing yes, no and unsure in this meta-analysis.

**Table 3 vaccines-10-01097-t003:** Studies examining factors related to individuals’ willingness to accept a first COVID-19 booster dose.

Reference	Older Age	Males	Married	Higher Educational Level	Ethnicity	Higher Income	Healthcare Workers	Residence	Chronic Comorbidities	History of COVID-19 Infection	Hospitalization	Higher Risk Perception of Infection	Higher Severity Perception of COVID-19
Lounis et al. [30]	↑	↑	NS	↓	-	-	↓	NS	↑	↑	NS	-	-
Rzymski et al. [34]	↑	↓	-	NS	-	-	-	NS	↑	↑	-	-	-
Lai et al. [29]	↓	NS	NS	↓	-	-	-	NS	NS	↑	-	NS	NS
Al-Qerem et al. [26]	NS	-	-	NS	-	-	-	NS	-	-	-	NS	NS
Wu et al. [37]	↑	-	-	-	-	-	↑	-	-	-	-	↑	↓
Miao et al. [31]	↑	↓	↑	↑	-	-	-	NS	↓	-	-	-	-
Chu et al. [27]	NS	NS	NS	↑	NS	NS	NS	NS	NS	NS	-	-	-
Jørgensen et al. [28]	↑	NS	-	NS	-	-	-	-	-	-	-	-	↑
Wang et al. [35]	↓	↑	-	↓	-	NS	↓	NS	-	-	-	-	-
Paul et al. [32]	↑	NS	NS	↑	NS	↑	NS	NS	↑	NS	-	↑	↑
Yoshida et al. [39]	↑	NS	-	-	-	-	-	-	-	-	-	-	-
Wirawan et al. [36]	↑	NS	-	↑	-	↑	-	-	-	NS	-	↑	↑
Positive association ^a^	8/12	2/10	1/5	4/10	0/2	2/4	1/5	0/8	3/6	3/6	0/1	3/5	3/6
Negative association ^b^	2/12	2/10	0/5	3/10	0/2	0/4	2/5	0/8	1/6	0/6	0/1	0/5	1/6
No association ^c^	2/12	6/10	4/5	3/10	2/2	2/4	2/5	8/8	2/6	3/6	1/1	2/5	2/6
**Reference**	**Infection in Family**		**Mortality in Family**		**Influenza Vaccine**		**Confidence in Healthcare System/Government/Physicians**		**Confidence in COVID-19 Vaccines**	**Confidence in COVID-19 Booster Doses**	**Compliance with Prevention Measures**	**Adverse Reactions and Discomfort Experienced after Previous Doses**	**Concerns for Serious Adverse Reactions to COVID-19 Booster Doses**
Lounis et al. [30]	NS		NS		NS		↑		↑	↑	-	-	-
Rzymski et al. [34]	-		-		↑		-		-	↑	-	↓	↓
Lai et al. [29]	-		-		-		-		-	↑	-	-	↓
Miao et al. [31]	-		-		-		↑		-	-	-	↓	-
Wang et al. [35]	-		-		↑		-		↑	-	-	-	-
Paul et al. [32]	-		-		-		NS		-	-	↑	-	-
Al-Qerem et al. [26]	-		-		-		-		-	-	-	↓	-
Wu et al. [37]	-		-		-		-		-	-	-	↓	-
Wirawan et al. [36]	NS		-		-		-		-	↑	-	-	-
Positive association ^a^	0/2		0/1		2/3		2/3		2/2	4/4	1/1	0/4	0/2
Negative association ^b^	0/2		0/1		0/3		0/3		0/2	0/4	0/1	4/4	2/2
No association ^c^	2/2		1/1		1/3		1/3		0/2	0/4	0/1	0/4	0/2
**Reference**	**Desire to Travel Abroad**		**Harm in Immune System**		**Further Vaccination Is Unnecessary**		**Low Safety of COVID-19 Booster Doses**			**Knowledge Level**		**Initial Uncertainty and Unwillingness to Accept the First COVID-19 Vaccine**	
Lounis et al. [30]	↑		↓		-		-			-		-	
Rzymski et al. [34]	-		-		↓		-			-		-	
Lai et al. [29]	-		-		-		↓			-		-	
Al-Qerem et al. [26]	-		-		-		-			NS		-	
Paul et al. [32]	-		-		-		-			NS		↓	
Positive association ^a^	1/1		0/1		0/1		0/1			0/2		0/1	
Negative association ^b^	0/1		1/1		1/1		1/1			0/2		1/1	
No association ^c^	0/1		0/1		0/1		0/1			2/2		0/1	

^a^ number of studies with a positive significant association (*p*-value < 0.05) between the predictor and individuals’ willingness to accept a first COVID-19 booster dose/total number of studies that examined the predictor. ^b^ number of studies with a negative significant association (*p*-value < 0.05) between the predictor and individuals’ willingness to accept a first COVID-19 booster dose/total number of studies that examined the predictor. ^c^ number of studies without a significant association (*p*-value ≥ 0.05) between the predictor and individuals’ willingness to accept a first COVID-19 booster dose/total number of studies that examined the predictor. NS: non-significant. ↑ more likely to accept. ↓ less likely to accept. - not investigated.

## Data Availability

The data presented in this study are available on request from the corresponding author.

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
