# Peer review of "First COVID-19 Booster Dose in the General Population: A Systematic Review and Meta-Analysis of Willingness and Its Predictors"

_vaccines, 2022, doi:10.3390/vaccines10071097_

Round 1

Reviewer 1 Report

The manuscript by Galanis and coworkers investigated the acceptance of first COVID-19 booster dose and its associated factors among fully vaccinated individuals. Moreover, the authors estimated the percentage of the fully vaccinated individuals that refused a first booster dose, and were unsure. Until now, no systematic review has investigated individuals’ intention to accept COVID-19 booster doses. Overall, this work is interesting and informative. Therefore, I recommend the manuscript be accepted by Vaccines after addressing the following issues in a minor revision.

11. What does the size of the black squares in Figure 2-4 stand for? How about the length of the lines in the Figure 2-4? The authors should clarify it in the caption or in the manuscript.

22. The authors should double check the format of the references. E.g., ref. 4, ref. 37, ref. 61.

33. There are two “ref. 1” in the reference section. The authors should correct it.  

Author Response

Dear Reviewer,

Thank you for giving us the opportunity to revise our manuscript entitled "Willingness, refusal and predictors of individuals to accept a first COVID-19 booster dose: a systematic review and meta-analysis". We would also like to thank you for your insightful comments and suggestions on how to improve our manuscript. We respectfully tried to address the issues raised and to revise our manuscript accordingly. We hope that our revision will reach the high standards of the Journal “Vaccines”. Please note that we explain how we addressed all issues brought up in your letter and to the specific points raised by the Reviewers, in the subsequent pages of our response letter. Also, we made some changes in the manuscript according to the other Reviewers’ instructions.

We look forward to hearing from you

Best Regards

The authors

The manuscript by Galanis and coworkers investigated the acceptance of first COVID-19 booster dose and its associated factors among fully vaccinated individuals. Moreover, the authors estimated the percentage of the fully vaccinated individuals that refused a first booster dose, and were unsure. Until now, no systematic review has investigated individuals’ intention to accept COVID-19 booster doses. Overall, this work is interesting and informative. Therefore, I recommend the manuscript be accepted by Vaccines after addressing the following issues in a minor revision.

  1. What does the size of the black squares in Figure 2-4 stand for? How about the length of the lines in the Figure 2-4? The authors should clarify it in the caption or in the manuscript.

Answer

Done. We add this information in the caption. Please, see the Figures 2-4.

  1. The authors should double check the format of the references. E.g., ref. 4, ref. 37, ref. 61.

Answer

Done. We check all the references.

  1. There are two “ref. 1” in the reference section. The authors should correct it.  

Done. We correct it.  

Reviewer 2 Report

The meta-analysis conducted by Petros et al. aimed to investigate the willingness and refusal of individuals to accept COVID-19 booster. The research project is designed appropriately and the methods were adequately described. I believe that the manuscript is well prepared and ready for publications. The topic of the research is significant for further efforts to prevent COVID19 pandemics.

There is just one suggestion for this research. Acceptance for vaccination might be strongly influenced by cultural background of study subjects. I have carefully checked figure 1 of this study and it seems for me a subgroup analysis might explain, at least partly, the heterogeneity of the results. The studies enrolled in the meta-analysis could be classified into two groups: countries belong to the Confucian culture circle (including China, Vietnam, and Japan) and the countries that do not belong to the Confucian culture circle (Jordan, Indonesia and the European countries). This subgroup analysis might obtain some results with very limited heterogeneity.

Author Response

Dear Reviewer,

Thank you for giving us the opportunity to revise our manuscript entitled "Willingness, refusal and predictors of individuals to accept a first COVID-19 booster dose: a systematic review and meta-analysis". We would also like to thank you for their insightful comments and suggestions on how to improve our manuscript. We respectfully tried to address the issues raised and to revise our manuscript accordingly. We hope that our revision will reach the high standards of the Journal “Vaccines”. Please note that we explain how we addressed all issues brought up in your letter and to the specific points raised by the Reviewers, in the subsequent pages of our response letter. Also, we made some changes in the manuscript according to the other Reviewers’ instructions.

We look forward to hearing from you

Best Regards

The authors

The meta-analysis conducted by Petros et al. aimed to investigate the willingness and refusal of individuals to accept COVID-19 booster. The research project is designed appropriately and the methods were adequately described. I believe that the manuscript is well prepared and ready for publications. The topic of the research is significant for further efforts to prevent COVID19 pandemics.

There is just one suggestion for this research. Acceptance for vaccination might be strongly influenced by cultural background of study subjects. I have carefully checked figure 1 of this study and it seems for me a subgroup analysis might explain, at least partly, the heterogeneity of the results. The studies enrolled in the meta-analysis could be classified into two groups: countries belong to the Confucian culture circle (including China, Vietnam, and Japan) and the countries that do not belong to the Confucian culture circle (Jordan, Indonesia and the European countries). This subgroup analysis might obtain some results with very limited heterogeneity.

Answer

Done. Dear Reviewer thank you very much for this inspiration. Although the heterogeneity among results remains high in the two groups (countries belong or not to the Confucian culture circle) we found an important difference in the willingness rate between the two groups (88.5% vs. 71.8%). We add the following text in the results and a respective comment in the Discussion section.

3.7. Impact of the countries

We separated countries according to the Confucian culture circle (China, Vietnam, and Japan) and we found that the prevalence of willingness was higher in studies that were conducted in countries that belong to the Confucian culture circle (88.5%, 95% CI=79.2-95.3%, I2=99.78%) than in studies that were conducted in countries that not belong to the Confucian culture circle (71.8%, 95% CI=57.5-84.1%, I2=99.89%). On the other hand, there was no impact of the countries on the individuals’ refusal of a COVID-19 booster dose. In particular, the prevalence of refusal in studies that were conducted in countries that belong to the Confucian culture circle was 7.8% (95% CI=1.3-19.2%, I2=99.72%), and in studies that were conducted in countries that not belong to the Confucian culture circle was 7.0% (95% CI=3.3-11.9%, I2=99.70%).

Reviewer 3 Report

A very interesting, educational and well written manuscript; however, there are some editing issues that the authors should consider and address.  The following are suggestions/comments for the authors.  Lines 19 & 20, "...acceptance of a first COVID-19 booster ...".  Lines 27 & 28, "...previous CoViD-19 vaccine doses and ...".  Line 29, "the burden of COVID-19, a high acceptance rate of booster doses could be critical in controlling the pandemic."  Line 44, "...disease decreased six months after ...".  Line 60, "...to investigate the acceptance of a first booster ...".  Line 104, "...the CoviD-19 booster vaccine?", etc.) and ...".  Line 122, "...unsure option), quality of the studies, and the ...".  Line 10, "...plot was asymmetrical (Supplementary ...".  Line 18, "...plot was asymmetrical (Supplementary ...".  Line 27, "...plot was asymmetrical (Supplementary ...".  Line 47, "Similarly, the impact of the ...".  Lines 69, "...accept further vaccination(s)."  Line 71, "...studies found no effect with age."  Line 77, "...likely to report an intention to accept a ...".  Lines 80 & 81, "...chronic comorbidities there was no association with willingness ...".  Line 83, "...2/2, 4/5, and 8/8 studies, respectively."  Line 91, "...was associated with an increased acceptance of a ...".  In the Discussion, line 13, "...on boosters and yet may be ...".  Line 17, "...but there is still a potential for improvement."  Lines 18 & 19, "...with fully vaccinated individuals, the acceptance rate of ...".  Line 28, "...communication strategies, emphasizing vaccination ...".  Line 31, "...we found that the willingness rate was ...".  Line 46, "Females with limited knowledge ...".  Line 48, "For example, male adolescents had ...".  Lines 49 & 50, "...female adolescents [44].  While females have lower vaccination rates...".  Line 61, "Many people discuss concerns about the safety, ...".  Lines 62 & 63, "...less willing to vaccinate.  Public concerns about ...".  Line 73, "...previously experienced side effects."  Line 75, "...long-term side effects of COVID-19 vaccinations [59]."  Line 77, "...health authorities to increase the vaccine ...".  Line 84, "...complications since an older age is a predictor ...".  Line 87, "...that people with a higher educational ...".  Line 88, "...literature, since an increased educational level ...".  Line 93, "...information sources [63].  An educational level ...".  Line 94, "...and awareness of individuals towards ...".  Lines 95 & 96, "...misinformation increases COVID-19 vaccine hesitancy, especially among ...".  Line 101, "achievements, since people whose incomes ...".  Line 117, "Sampling methods may influence ...".  Line 137, "...on booster doses.  With a ...".  Line 149, "...towards COVID-19 booster doses, providing community ...".  

Author Response

Dear Reviewer,

Thank you for giving us the opportunity to revise our manuscript entitled "Willingness, refusal and predictors of individuals to accept a first COVID-19 booster dose: a systematic review and meta-analysis". We would also like to thank you for their insightful comments and suggestions on how to improve our manuscript. We respectfully tried to address the issues raised and to revise our manuscript accordingly. We hope that our revision will reach the high standards of the Journal “Vaccines”. Please note that we explain how we addressed all issues brought up in your letter and to the specific points raised by the Reviewers, in the subsequent pages of our response letter. Also, we made some changes in the manuscript according to the other Reviewers’ instructions.

We look forward to hearing from you

Best Regards

The authors

A very interesting, educational and well written manuscript; however, there are some editing issues that the authors should consider and address.  The following are suggestions/comments for the authors. 

Lines 19 & 20, "...acceptance of a first COVID-19 booster ...". 

Lines 27 & 28, "...previous CoViD-19 vaccine doses and ...". 

Line 29, "the burden of COVID-19, a high acceptance rate of booster doses could be critical in controlling the pandemic." 

Line 44, "...disease decreased six months after ...". 

Line 60, "...to investigate the acceptance of a first booster ...". 

Line 104, "...the CoviD-19 booster vaccine?", etc.) and ...". 

Line 122, "...unsure option), quality of the studies, and the ...". 

Line 10, "...plot was asymmetrical (Supplementary ...". 

Line 18, "...plot was asymmetrical (Supplementary ...". 

Line 27, "...plot was asymmetrical (Supplementary ...". 

Line 47, "Similarly, the impact of the ...". 

Lines 69, "...accept further vaccination(s)." 

Line 71, "...studies found no effect with age." 

Line 77, "...likely to report an intention to accept a ...". 

Lines 80 & 81, "...chronic comorbidities there was no association with willingness ...". 

Line 83, "...2/2, 4/5, and 8/8 studies, respectively." 

Line 91, "...was associated with an increased acceptance of a ...". 

In the Discussion, line 13, "...on boosters and yet may be ...". 

Line 17, "...but there is still a potential for improvement." 

Lines 18 & 19, "...with fully vaccinated individuals, the acceptance rate of ...". 

Line 28, "...communication strategies, emphasizing vaccination ...". 

Line 31, "...we found that the willingness rate was ...". 

Line 46, "Females with limited knowledge ...". 

Line 48, "For example, male adolescents had ...". 

Lines 49 & 50, "...female adolescents [44].  While females have lower vaccination rates...". 

Line 61, "Many people discuss concerns about the safety, ...". 

Lines 62 & 63, "...less willing to vaccinate.  Public concerns about ...". 

Line 73, "...previously experienced side effects." 

Line 75, "...long-term side effects of COVID-19 vaccinations [59]." 

Line 77, "...health authorities to increase the vaccine ...". 

Line 84, "...complications since an older age is a predictor ...". 

Line 87, "...that people with a higher educational ...". 

Line 88, "...literature, since an increased educational level ...". 

Line 93, "...information sources [63].  An educational level ...". 

Line 94, "...and awareness of individuals towards ...". 

Lines 95 & 96, "...misinformation increases COVID-19 vaccine hesitancy, especially among ...". 

Line 101, "achievements, since people whose incomes ...". 

Line 117, "Sampling methods may influence ...". 

Line 137, "...on booster doses.  With a ...". 

Line 149, "...towards COVID-19 booster doses, providing community ...".  

Answer

Done. Dear Reviewer thank you very much for your suggestions. We make the respective changes in our manuscript.